# The Impact of Metformin on Tumor-Infiltrated Immune Cells: Preclinical and Clinical Studies

**DOI:** 10.3390/ijms241713353

**Published:** 2023-08-28

**Authors:** Mohamed Abdelmoneim, Mona Alhussein Aboalela, Yoshinori Naoe, Shigeru Matsumura, Ibrahim Ragab Eissa, Itzel Bustos-Villalobos, Patricia Angela Sibal, Yuhei Takido, Yasuhiro Kodera, Hideki Kasuya

**Affiliations:** 1Department of Surgery II, Graduate School of Medicine, Nagoya University, Nagoya 466-8550, Japan; montipedia@yahoo.com (M.A.A.); brahim_essa@yahoo.com (I.R.E.);; 2Cancer Immune Therapy Research Center, Graduate School of Medicine, Nagoya University, Nagoya 466-8550, Japansmatsumu@med.nagoya-u.ac.jp (S.M.);; 3Department of Neurosurgery, Graduate School of Medicine, Nagoya University, Nagoya 466-8550, Japan

**Keywords:** tumor microenvironment, tumor-infiltrating lymphocytes, metformin

## Abstract

The tumor microenvironment (TME) plays a pivotal role in the fate of cancer cells, and tumor-infiltrating immune cells have emerged as key players in shaping this complex milieu. Cancer is one of the leading causes of death in the world. The most common standard treatments for cancer are surgery, radiation therapy, and chemotherapeutic drugs. In the last decade, immunotherapy has had a potential effect on the treatment of cancer patients with poor prognoses. One of the immune therapeutic targeted approaches that shows anticancer efficacy is a type 2 diabetes medication, metformin. Beyond its glycemic control properties, studies have revealed intriguing immunomodulatory properties of metformin. Meanwhile, several studies focus on the impact of metformin on tumor-infiltrating immune cells in various tumor models. In several tumor models, metformin can modulate tumor-infiltrated effector immune cells, CD8^+^, CD4^+^ T cells, and natural killer (NK) cells, as well as suppressor immune cells, T regulatory cells, tumor-associated macrophages (TAMs), and myeloid-derived suppressor cells (MDSCs). In this review, we discuss the role of metformin in modulating tumor-infiltrating immune cells in different preclinical models and clinical trials. Both preclinical and clinical studies suggest that metformin holds promise as adjunctive therapy in cancer treatment by modulating the immune response within the tumor microenvironment. Nonetheless, both the tumor type and the combined therapy have an impact on the specific targets of metformin in the TME. Further investigations are warranted to elucidate the precise mechanisms underlying the immunomodulatory effects of metformin and to optimize its clinical application in cancer patients.

## 1. Introduction

Cancer is a significant global health challenge around the world, with a very high incidence and mortality rate. The incidence and mortality of cancer increased over the last few decades; there were an estimated 19.3 million new cancer cases and 10 million cancer-related deaths in 2020 [1]. The current standard of care for cancer patients varies depending on the type and stage of cancer, as well as individual patient characteristics. To achieve the best results for each patient, the most effective and personalized treatment plan must be provided [2].

Immunotherapy has emerged as a groundbreaking approach for cancer treatment. It aims to harness the body’s immune system to recognize and destroy cancer cells. Checkpoint inhibitors, such as programmed cell death protein 1 (PD-1) and cytotoxic T-lymphocyte-associated protein 4 (CTLA-4) inhibitors, have shown remarkable success in treating various cancers [3,4]. However, the prognosis of cancer immunotherapy is varied between treated patients. One cause of these treatments’ limitations is the complexity of the tumor microenvironment (TME) and its components, including tumor-infiltrating immune cells. Moreover, tumor vascularization reduces the efficiency of immunotherapy since it plays a crucial role in tumor-infiltrating immune cells [5].

The metabolism of cells plays a significant role in shaping TME and its immunosuppressive characteristics. Tumor cells exhibit unique metabolic adaptations that can directly or indirectly impact the immune system’s response. However, the immune-infiltrated immune cells may undergo metabolic reprogramming to adapt to the nutrient-deprived and immunosuppressive conditions [6]. Therefore, understanding the tumor-infiltrating immune cells is essential to the fate of tumor therapy [7].

Metformin is an FDA-approved drug to treat type 2 diabetes mellitus (T2DM), and it is known as 1,1-dimethyl biguanide hydrochloride. Metformin is a biguanide derived from an herbaceous plant called Galega officinalis (known as goat’s rue), an herbal medicine in Europe found to be rich in guanidine [8]. In 1957, French physician Jean Strene published about using metformin as an antidiabetic drug. However, metformin attracted minimal attention because it showed a weaker effect than other biguanides, such as buformin and phenformin. Afterward, buformin and phenformin were withdrawn in the 1970s due to lactic acidosis. After then, metformin was introduced in the USA and approved in 1995, which encouraged additional research and clinical use. As a result, metformin has now become the most often prescribed antidiabetic medicine in the world [8].

In 2005, a study describing a decrease in cancer burden in T2DM patients treated with metformin as compared to those treated with alternative diabetic therapies sparked interest in the potential relevance of biguanides in cancer disease [9]. Moreover, several epidemiological studies showed that metformin has anticancer effects besides its glycemic control in T2DM, which enhances the need to understand the anticancer mechanisms of this drug [10,11]. Metformin treatment has been shown to decrease the rate of cancer and/or improve the prognosis of diabetic patients with pancreatic, hepatic, colon, breast, and bladder cancer [10]. Therefore, research on metformin as an anticancer drug has recently received more interest. 

Metformin is one of the small molecule modulators that can target the PI3K/AKT/mTOR signaling pathway for immune cells and/or cancer cells [12]. In tumor cells, metformin inhibits complex I of the electron transport chain (ETC), which leads to NADH accumulation inside tumor cells and diminishes ATP production inside the cells. Decreased ATP levels led to the activation of AMP-activated protein kinase (AMPK), which activates the AMPK pathway and inhibits the mammalian target of rapamycin (mTOR), therefore enhancing tumor cell death [13]. Tumor cell death induced by metformin supports the expansion of tumor-infiltrating lymphocytes (TILs) [14,15]. Interestingly, metformin can also restore chemotherapy sensitivity by modulating NF-kB, ERK1/2 activation, autophagy, and the population of cancer stem cells [16]. However, the mechanism of action that clarifies metformin’s effect on immune cells in the TME is not fully understood. In this review, we summarize preclinical and clinical studies that studied the impact of metformin, either alone or combined, on tumor-infiltrating immune cells and its targeted pathway. 

## 2. Effect of Metformin on CD8^+^ TILs

TILs have drawn a lot of attention due to their role in cancer prognosis. TILs include CD8^+^ T cells, CD4^+^ T cells, B cells, NK cells, and regulatory T cells [17]. CD8^+^ T cells play an essential role in antitumor immune response [18]. High CD8^+^ T cell infiltration improves prognosis and survival in cancer patients [19]. In several mouse models, including those for leukemia, melanoma, colon cancer, fibrosarcoma, and triple-negative breast cancer (TNBC), metformin therapy had a favorable impact on the tumor-infiltrated CD8^+^ T cells, such as an increased number of CD8^+^ TILs [20,21,22,23,24,25]. Besides preclinical murine models, metformin has been investigated in various clinical trials to determine its effects on CD8^+^ TILs in cancer patients. In diabetic patients with lung cancer, metformin treatment increases the frequencies of central memory and memory stem CD8^+^ TILs. Also, metformin treatment on patients with head and neck squamous cell carcinoma (HNSCC), esophageal squamous cell carcinoma (ESCC), colorectal cancer (CRC), or breast cancer has shown the enhanced infiltration of cytotoxic CD8^+^ TILs [26,27,28,29,30]. Moreover, metformin treatment led to increased expression of IFNγ and IL-2 in the surgical specimens of treated patients with early breast cancer [30]. However, metformin combined with platinum-based chemotherapy drugs does not increase CD8^+^ cytotoxic T cells in the stroma of the tumor in patients with advanced-stage epithelial ovarian cancer (EOC) [31]. Overall, these preclinical and clinical trials suggest that metformin treatment has positive effects on CD8^+^ TILs in various cancer types. Although with those accumulated observations, metformin has been establishing its antitumor effects on CD8+ TILs, its mechanism of action has still not yet been fully elucidated.

In a leukemia and melanoma mouse models, metformin treatment increases the infiltration of CD8^+^ TILs, including both effector and effector memory CD8^+^ TILs, accompanied with an increase in the cytokine production, including IL-2, IFNγ, and TNFα [20,21,23]. Metformin treatment elevates AMPK (AMP-activated protein kinase) and Bat3 expression while inhibiting caspase-3 activity, thereby preventing apoptosis of CD8^+^ TILs [20]. In another study using a melanoma and fibrosarcoma mouse models, metformin treatment increases mitochondrial reactive oxygen species (mtROS) production in CD8^+^ TILs, which enhances Glut-1 cell surface expression and activates Nrf2 in CD8^+^ TILs. Nrf2, in turn, promotes selective autophagy and glutaminolysis, leading to the activation of mTORC1 and increased proliferation of CD8^+^ TILs [24].

## 3. Effect of Metformin on the PD1-PD-L1 Axis

Prolonged T cell stimulation induces T cell exhaustion that is mediated by programmed cell death 1 (PD-1) inhibitory receptor on T-cells. The interaction between PD-1/PD-L1 reduces CD8^+^ T cell activity and inhibits cytokine production and cell proliferation. Metformin treatment significantly decreases infiltration of exhausted CD8^+^ PD-1^+^ in several cancer models and, consequently, enhances CD8^+^ T cell activity [32,33]. The effects on CD8^+^ TILs were associated with the C-Jun-N-terminal Kinase (JNK) pathway [33]. On the other hand, metformin can indirectly activate CD8^+^ TILs by targeting tumor cells. In the breast cancer tumor model, metformin activates AMPK in tumor cells, which directly binds to and phosphorylates PD-L1 at S195, inducing abnormal PD-L1 glycosylation and leading to degradation of PD-L1 by ER-associated protein degradation (ERAD). Consequently, by decreasing the expression level of PD-L1 on the tumor cell surface, metformin treatment increases the activity of CD8^+^ TILs that are indicated by increased granzyme B production [34]. In another report, in a mouse ovarian tumor model, metformin suppressed STAT3 and c-Myc through activation of FOXO3 tumor suppressor protein in the tumor and subsequently downregulates PD-L1 expression, which leads to activation of CD8^+^ TILs [35]. Interestingly, metformin single treatment significantly decreases PDL-1 expression in a TNBC mouse model [25]. In a clinical trial of diabetic patients with non-small cell lung cancer (NSCLC), metformin increased the frequencies of central memory and memory stem CD8^+^ TILs. The effects were associated with the activation of AMPK, which downregulates microRNA-107 and enhances the expression of Eomesodermin (Eomes), an important transcription factor for memory CD8^+^ T cell formation. As a result, the transcription of the PDCD1 gene, encoding PD-1, is inhibited [36].

## 4. Combination Therapy with Metformin 

Therefore, combination therapies of metformin with immune checkpoint inhibitors (ICIs) would enhance the antitumor effects efficiently. Indeed, metformin combined with anti-PD-1 led to a significant increase in CD8^+^ TILs accompanied by a significant increase in granzyme B production in a melanoma mouse model [37]. Moreover, metformin combined with anti-PD1 antibody showed more CD8^+^ TILs infiltration and proliferation in both melanoma and TNBC mouse models [24,25]. In a syngeneic mouse model of human papillomavirus-associated head and neck cancer (mEER/MTEC), acute metformin exposure (40 mg/kg for five consecutive days) in a well-established tumor combined with anti-PD1 increases CD8^+^ TILs infiltration and proliferation [38]. Since metformin combined with nivolumab showed tumor regression in several mouse models, a phase Ib clinical trial has been started to evaluate the combination treatment through two stages. Stage 1 is evaluating the recommended dose of combination therapy in patients with refractory or recurrent solid tumors. However, in stage 2, combinations will be evaluated in patients with advanced or recurrent NSCLC, advanced or recurrent thymic epithelial tumors, and advanced or recurrent pancreatic cancer [39].

Aside from ICIs, metformin, in combination with various anticancer therapies, showed promising effects on the infiltrated CD8^+^ TILs, e.g., metformin combined with an oncolytic HSV-1 virus, C-REV, enhanced the infiltration of CD8^+^ TILs accompanied by a significant increase in IFNγ production in bilateral Pan02 tumor-bearing mice [40]. CD44^+^ and CD69^+^ expressing CD8^+^ TILs are effector cells that have an antitumor immune response [41,42]. Combination therapy significantly increases both effector CD44^+^ CD8^+^ PD-1^−^ and CD69^+^ CD8^+^ PD-1^−^ on both tumor sides, indicating the high activity of the infiltrating CD8^+^ TILs. Metformin single treatment did not significantly increase CD8^+^ TILs in Pan02 tumor model; hence, combination therapy was required to increase the effectiveness of the treatment [40]. In another 4T1 murine tumor model, metformin combined with a tumor membrane vesicles (TMVs) vaccine enhances CD8^+^ TILs infiltration, accompanied by a significant increase in central memory CD8^+^ T cells, CD8^+^ IFNγ^+^ TNF-α^+^, and CD8^+^ IFNγ^+^ IL-2^+^ compared to control group indicating the activity of CD8^+^ T cells. The expressions of the Ki-67 proliferation marker and terminal effector marker Tim-3 on CD8^+^ T cells are significantly increased in the combination group. Similar to the Pan02 tumor model, metformin single treatment did not significantly enhance central memory CD8^+^ T cells infiltration or the percentage of cytokine production [43]. Metformin monotherapy did not demonstrate a beneficial effect on CD8^+^ TILs in those tumor models, implying that the tumor type-dependent tumor microenvironment would influence the effects of metformin. In such cases, probably, the appropriate combination therapy would sensitize the tumor microenvironment to metformin. In conclusion, most preclinical studies that studied the effect of metformin on CD8^+^ TILs either, single or combined, showed favorable effect, as shown in Table 1.

## 5. Effect of Metformin on CD4^+^ T Helper Cells

CD4^+^ T helper (TH) cells play a crucial role in regulating the immune microenvironment within tumors. TH subtypes have varied functions that can either enhance or suppress immune responses to tumors [44]. Metformin treatment enhances infiltration of CD4^+^ TILs cells accompanied with a decrease in exhausted CD4^+^ PD-1^+^ TILs in a TNBC mouse model [33]. This finding was consistent with metformin treatment in patients with early breast cancer, and metformin treatment led to increased expression of CD4^+^ in the surgical specimens of the treated patients [30]. Moreover, metformin combined with anti-PD1 increases CD4^+^ TILs infiltration and proliferation in a mEER/MTEC mouse model [38]. However, in EOC patients, metformin combined with platinum-based chemotherapy drugs does not enhance CD4^+^ cytotoxic T cells in the stroma of the tumor [31].

In the TME, TH1 cells can secrete cytokines, e.g., IL-2, TNF-α and IFNγ, which enhance cancer cell death, as well as activate cytotoxic immune cells such as CD8^+^ and NK cells [45]. Interestingly, metformin increases CD4^+^ IFNγ^+^ TH1 cells percentage in CT-26 murine colon cancer model, which may enhance the antitumor immunity [22]. However, chronic exposure of metformin (40 mg/kg; 5 days/week) for 4 weeks prior to tumor inoculation, decreases TH1 cells infiltration into the tumor [38]. By contrast, TH17 cells worsen tumor growth and metastasis by impairing antitumor immunity and assisting survival of tumor cells [46]. Metformin has been demonstrated to prevent the differentiation of CD4^+^ T cells into TH17 cells by decreasing glucose transporter Glut-1 expression in naïve CD4^+^ T cells. Moreover, it decreases the production of IL-22 by TH17 and TH1 cells in a murine hepatocellular carcinoma (HCC) tumor model in a dose-dependent manner [47]. Another experimental study showed that metformin reduces IL-17A expression from TH17 in B16F10 melanoma mouse mice. This effect contributes to activation of Situnin 1 (SIRT1) pathway in CD4^+^ T cells, which reduces TH17 polarization [48]. Furthermore, metformin decreases the anti-metastatic activity by decreasing number of lung ROR-γ^+^IL-17A^+^CD4^+^ TH17 cells in a B16F10-challenged mouse model [23]. Additional studies are required to investigate the effect of metformin therapy on TH subsets infiltrated into tumor and its targeted pathway using various tumor models.

## 6. Effect of Metformin on Treg Cells

CD4^+^ CD25^+^ T regulatory cells (Treg) are immunosuppressive cells that enhance tumor growth, and they are considered one of the main immunotherapeutic targets for cancer [49]. Several studies assessed the efficacy of metformin on tumor-infiltrated Treg cells either alone or combined. Several preclinical studies reported a favorable impact of metformin in tumor-Treg using various mouse tumor models. In MethA tumor-bearing mice, metformin treatment decreased tumor-infiltrated Treg cells, especially terminally differentiated CD103^+^ KLRG-1^+^ Treg cells (active effector Treg cells) accompanied with a decrease in the inhibitory molecules expressed on them. They showed that pretreatment of naïve CD4^+^ T cells by metformin inhibits TGF-ß-dependent inducible Treg differentiation through mTORC1 activation, which inhibits the expression of FOXP3 [50]. Similarly, in the murine lung tumorigenesis model, metformin treatment decreases tumor-associated FOXP3^+^ Treg cells due to activation of AMPK, which inhibits the mTOR pathway [51]. Also, in another mouse tumor model, metformin significantly decreased the intratumoral Treg cells, accompanied with a reduction in local and metastatic tumor progression [52]. Furthermore, metformin combined with anti-PD1 or conjugated with self-delivery nanoparticles (MA-pepA-Ce6 NPs) decreases Treg cells, with a reduction of IL-10 cytokine production [53]. In a clinical trial of HNSCC patients, metformin treatment was associated with a 41.4% decrease in intratumoral FOXP3^+^ T cells in the biopsy [26]. Moreover, metformin combined with platinum-based chemotherapy drugs significantly decreases CD4^+^ FOXP3^+^ Treg and CD4^+^/FOXP3^−^ T helper cells in the tumor stroma of patients with advanced-stage EOC [31].

On the other hand, metformin combination with C-REV does not decrease terminally differentiated CD103^+^ KLRG-1^+^ T regulatory cells in both the injected and contralateral tumor sides compared to the single-treated groups in bilateral Pan02 tumor model [40]. In phase II clinical trials of ESCC patients, low-dose metformin did not affect the infiltration of CD4^+^ and FOXP3^+^ T cells [28].

## 7. Effect of Metformin on B Cells

Studies of metformin on tumor-infiltrated B cells are very limited. Metformin combined with platinum-based chemotherapy drugs does not affect CD20^+^ B cells neither in the stroma nor in the tumor islet in patients with advanced-stage EOC [31]. Future studies of metformin in various B cells subsets in the TME will be interesting.

## 8. Effect of Metformin on γδ T Cells

γδ T cells are a small subgroup of T cells with an antitumor function, and their presence in the TME is associated with favorable outcomes [54]. The impact of metformin on infiltrated γδ T cells in brain tumors was assessed in GL261 murine model. Metformin treatment enhances the infiltration of tumor-reactive γδ T cells with high cytotoxic activity. Metformin treatment reduces hypoxia-inducible factor-1a (Hif1a) and apoptotic marker Bax expression in infiltrated γδ T cells; however, it enhances NKG2D expression on the infiltrated cells through the cAMP-PKA axis [55].

## 9. Effect of Metformin on NK Cells

NK cells have an antitumor effect, and their presence in the TME correlates with the prognosis of several types of cancer [56]. Several studies reported the effect of metformin in tumor-infiltrated NK cells either alone or combined. Metformin treatment increases NKp46^+^ populations in the tumor tissue of ovarian tumor-bearing mice and, therefore, suppresses tumor growth [35]. Moreover, chronic metformin exposure increases NK cell infiltration in the mEER/MTEC syngeneic mouse model [38]. Metformin combined with anti-PD-1 led to a 6.4-fold increase in NK cells compared to the anti-PD1 single treatment group, while metformin single treatment led to a 3.9-fold increase in NK cells compared to the control group in B16F10 tumor-bearing mice. The activation of NK cells was mediated by the activation of p38 mitogen-activated protein kinases (MAPK), which has cytolytic granules secreting ability [37]. Additionally, metformin combined with local radiation therapy markedly increased NKp46/CD335^+^ NK cells in the metastatic tumors in a metastatic mouse lung model [57]. In the clinical trial of HNSCC patients, pretreatment of patients with metformin has been shown to significantly increase the presence of NKp46^+^ cells in the tumor tissue. Furthermore, in the coculture of tumor-infiltrating NK cells with established HNSCC cell lines, NK cells from metformin-pretreated tumors demonstrated higher cytotoxic activity in the tumor cells, suggesting that metformin enhances the cytotoxic function of NK cells. This effect of metformin treatment was achieved by the inhibition of the CXCL1 pathway and stimulation of the STAT1 pathway, specifically in NK cells. They have also shown that the enhanced NK cell cytotoxicity is independent of AMPK but is dependent on the mTOR and pSTAT1 pathways [58]. Additionally, metformin showed to promote antitumor activity of Nk cells through increased expression of immunostimulatory microRNA, miRNA-150 and miRNA-155; however, it suppressed immunosuppressive miRNA-146a expression [59].

## 10. Effect of Metformin on Myeloid-Derived Suppressor Cells (MDSCs)

MDSCs are immunosuppressive cells that are well distinguished from other myeloid cells in the TME [60]. Several studies evaluated the effect of metformin on MDSCs either alone or combined. In the osteosarcoma mice model and murine colon cancer tumor model, metformin-mediated tumor growth inhibition through decreased polymorphonuclear PMN-MDSCs CD11b^+^ Ly6C^mid^ Ly6G^high^ and CD11b^+^ Gr-1^+^ MDSCs in tumors [22,61]. Additionally, the uptake assay of 2-NBDG, a traceable glucose analog, showed a decrease in MDSCs, suggesting that metformin confers a lower energetic quiescent state to MDSCs. They showed that metformin modulates the metabolism of CD11b^+^ cells by elevating glycolysis and lowering oxidative phosphorylation [61]. Another study has shown that metformin inhibits MDSC in an AMPK/STAT3-dependent manner [22]. In addition, metformin suppresses intraperitoneal tumor growth through a decreased percentage of CD11b^+^ Gr-1^+^ MDSCs in the peritoneal lavage fluid of the peritoneal dissemination mouse model, which is induced by intraperitoneal injection of RLmale1 [62]. Furthermore, in 4T1 breast cancer mouse model, metformin treatment significantly suppress CD11b^+^ Gr-1^+^ MDSCs [63]. Intraperitoneal acute metformin exposure also decreased Ly6G^+^ G-MDSCs in mEER/MTEC syngeneic mouse model [38]. Metformin combined with local radiation therapy significantly decreases Gr-1^+^ cells (MDSCs) in the lung metastatic lesion [57]. Metformin combined with a TMVs tumor vaccine significantly reduces CD11b^+^ Ly6G^+^ MDSCs in murine oral carcinoma-2 (MOC2) murine tumor model [43]. These findings were consistent with metformin treatment of ovarian cancer patients. Metformin suppressed the function of MDSCs via inhibition of CD39 and CD73 ectoenzymatic activity on both M-MDSCs and PMN-MDSCs through downregulation of hypoxia-inducible factor 1 (HIF-1), which is mediated by AMPK activation [64]. Moreover, metformin treatment reduces PMN-MDSCs accumulation in diabetic patients with ESCC tumors, accompanied by a significant reduction in myeloid-specific receptor, CD33 expression, and iNOS expression associated with PMN-MDSC function. Metformin mediates the PMN-MDSCs reduction in the tumor via activation of AMPK, which upregulates dachshund homolog 1 (DACH1) expression on tumor cells and, therefore, downregulates CXCL1 expression [65].

## 11. Effect of Metformin on Tumor-Associated Macrophages (TAMs)

TAMs are myeloid cells that establish an immunosuppressive TME and support tumor growth [66]. Since TAMs have a negative impact on TME, several studies investigated the efficacy of metformin on them. Metformin enhances the shift from an M2 to an M1-like phenotype accompanied by a significant reduction in reactive oxygen species (ROS) and glucose uptake in CD11b^+^ Gr-1^low^ F4/80^high^ TAMs, enhancing tumor inhibition [61,67]. In the MOC1 mouse tumor model, metformin combined with a TMVs tumor vaccine significantly increases the expression of CD86 and MHC-II activation markers in TAMs but without a reduction in PD-L1 expression [43]. Consistent with a clinical trial of CRC patients, metformin treatment significantly decreased the rate of M2 macrophages in the tumor defined by CD163^+^/ CD68^+^ expression. Therefore, metformin switches the balance of TAM from M2 to M1, which may suppress tumor progression and enhance the patient’s prognosis [29]. In advanced EOC patients, metformin treatment combined with platinum-based chemotherapy drugs reduces CD68^+^ macrophages in the tumor, however, with no statistical difference [31]. Interestingly, in phase II clinical trial on ESCC patients, a low dose of metformin does not increase the total number of F4/80^+^ macrophage populations; however, it decreases tumor-promoting CD163^+^ macrophages and increases tumor-suppressive CD11c^+^ macrophages [28]. Overall, these findings suggest that metformin has the potential to shift the balance of TAMs from an immunosuppressive M2 phenotype to an antitumor M1 phenotype.

## 12. Effect of Metformin on Tumor-Infiltrated Dendritic Cells (DCs)

Dendritic cells are antigen-presenting cells, which are essential for antigen presentation and activation of CD8^+^ T cells in tumor-drained lymph nodes (TDLNs) [68]. CD8^+^ T cells that infiltrated into the tumor are co-stimulated by DCs to drive their effector differentiation [69]. Therefore, targeting DCs in tumors by metformin is essential to be investigated. Chronic metformin exposure increases DCs infiltration by two folds in the mEER syngeneic mouse model, which supports antitumor activity [38]. Moreover, in the murine melanoma model, RNA sequencing analysis from sorted CD8^+^ T cells showed that metformin treatment increases the gene expression of CD8^+^ DCs. Additionally, metformin combined with anti-PD-1 antibody treatment enhances the CD11c^+^ CD11b^−^ DC population in the tumor, thus enhancing antitumor immunity [24]. Combining metformin with other immunotherapeutic approaches, such as tumor vaccines or oncolytic viruses, has shown positive effects on DC activity. In the MOC1 and MOC2 tumor models, the combination of metformin with a tumor vaccine increased the expression of CD86 and MHC-II, indicating an activated status of DCs [43]. Similarly, in combination with C-REV oncolytic virus, metformin efficiently modulated conventional dendritic cells type-1 (cDC1) in both tumors and TDLNs. These cDC1s exhibited increased MHC-I expression and enhanced activity, as indicated by high XCR-1 expression, which is associated with efficient cross-presentation of antigens [40]. Interestingly, in 4T1 breast cancer mouse model, metformin treatment significantly increases CD11c^+^ F4/80^−^ CD11b^+^ B220^−^ cDCs and in CD11c^+^ F4/80^−^ CD11b^−^ B220^+^ pDCs [63]. The findings suggest that metformin has the potential to enhance the activity and infiltration of DCs in the TME, which can contribute to improved antitumor immunity. Furthermore, metformin’s effects on DCs could be examined in clinical trials that evaluate metformin monotherapy or combination therapy, and the pathway involved in DC activation should be elucidated in the future.

## 13. Summary and Future Directions

The immune profile in the TME plays a crucial role in shaping the immune response and determining the outcome of cancer therapy. Here, we discussed the modulation of infiltrated immune cells in the TME by metformin. Metformin has been found to have a favorable effect on CD8^+^ TILs, especially in hot tumors. Metformin has the potential to upregulate tumor-targeting immune cells and to downregulate tumor-supportive immune cells at the same time. However, we speculate that metformin would have marginal effects on cold tumors, which are mainly composed of immunosuppressive cells. The effects of metformin on each population seem not strong enough to change the TME of the cold tumor. Therefore, metformin combination treatments, including Ovs and/or ICIs, would be considered a great candidate for cold tumor treatment.

On the other side, the metformin targets on immune cells are varied from one tumor model to another. In several tumor models, although metformin can hardly modulate CD8^+^ TILs, it can modulate one of the immune-suppressive cells. Therefore, understanding the pathways targeted by metformin in different immune cells is important.

Generally, metformin can modulate the immune cells in the TME either in AMPK-dependent or AMPK-independent manner (Figure 1). Metformin activates AMPK and inhibits mTOR in CD8^+^ TILs, and increases their cytokine production, enhancing antitumor immunity [20,21]. Activation of AMPK also reduces microRNA-107 expression and enhances the expression Eomes, leading to PD-1 inhibition [36]. On the other hand, metformin activates Nrf2/mTORC1/p62 axis in CD8^+^ TILs via stimulation of mtROS production in an AMPK-independent manner, which enhance CD8^+^ TILs proliferation and IFNγ production [24]. Metformin also modulates the PD-1/PDL-1 axis by targeting tumor cells. Metformin mediates dual blocking of STAT3-PDL-1 and c-Myc-PDL1 pathways in tumor through upregulation of FOXO3, and therefore, it enhances CD8^+^ TILs activation [35]. Furthermore, metformin modulates glucose metabolism in naïve CD4^+^ T cells by reducing Glut-1 expression and, therefore, restrains their differentiation into TH17 cells. This effect is mediated by downregulation of the phosphorylation of STAT3 and STAT4 through targeting AMPK/mTOR pathway [47]. This finding was consistent with another study, which showed that metformin activates SIRT1 pathway in TH17 cells that deacetylates STAT3, thus reducing its ability to bind to Rorc (retinoic acid receptor-related orphan receptor) promoter and, therefore, impedes their differentiation [48].

In tumor-infiltrated Treg cells, metformin inhibits FOXP3 expression in an AMPK-dependent manner through the activation of mTORC1, which acts as a negative regulator for FOXP3 induction in naive CD4^+^ T cells [50]. In addition, metformin activates NK cells infiltrated to the tumor accompanied with high perforin production through mTOR inhibition that subsequently suppresses CXCL1 production in an AMPK-independent manner but is dependent on the mTOR and pSTAT1 pathways [58]. In some tumor models, metformin showed antitumor efficacy based on the reduction of CD11b^+^ myeloid cells (MDSCs and TAMs) in the tumor [22,61,62,63,64,65,66]. Metformin inhibits CD39 and CD73 ectoenzymatic activity on MDSCs through downregulation HIF-1 in an AMPK-dependent manner, which led to MDSCs suppression [63]. Metformin also reduces MDSCs accumulation through upregulation of DACH1, which downregulates CXCL1 expression in an AMPK-dependent manner [65]. Moreover, metformin showed to modulate macrophage polarization from M2 to M1 through activation of AMPK-NF-KB signaling in tumor and thus suppress tumor growth [67]. However, metformin showed to enhance the shift from an M2 to an M1 in an AMPK-independent manner, through reduction in CD206 as well as ROS and glucose uptake in TAMs [61].

Thus, Metformin possesses the ability to influence various interactions within the TME. Metformin enhances CD8^+^ T, γδ T, and NK cells infiltration in several tumor models and activates AMPK, which plays a pivotal role in quelling immune checkpoints and enhancing the release of antitumor cytokines. Additionally, it has the capacity to stimulate TH1 responses while constraining TH2 cell multiplication, resulting in an augmented release of cytokines that combat tumors. Interestingly, metformin shows the potential to enhance DCs and cDC1 subset that enhances cross-presentation of tumor antigens into CD8^+^ TILs. Moreover, it can restore exhausted CD8^+^ TILs, which may boost antitumor immunity, and increase the response to ICIs. On the other hand, metformin can counteract the immunosuppressive effects caused by Tregs, MDSCs, and TAMs as well as attenuate the release of their cytokines. These prevent supportive effects for cancer cells, decrease angiogenesis, and allow CD8^+^ T and NK cells to attack cancer cells. Furthermore, the inhibition of growth factors facilitates the induction of apoptosis in cancer cells.

## 14. Strategies to Enhance the Clinical Utility of Metformin

Maximizing metformin’s potential benefits while minimizing any potential risk is necessary to increase its clinical usefulness. Here, we listed various clinical studies that investigated the role of metformin on tumor-infiltrating immune cells in various types of cancer in Table 2.

It is well known that in vitro high concentrations of metformin are necessary to inhibit cancer cell growth via impairing ATP synthesis. However, in vivo high plasma concentration of metformin that impair ATP synthesis cannot be achieved in patients treated with the standard clinical doses of metformin. In the tumor tissue from mouse models, oral metformin treatment led to a low concentration (micromolar range) [70]. Understanding the reason that very low concentrations of metformin cannot affect tumor cells in vitro but it affects them in vivo is interesting. Indeed, the in vivo and in vitro results of metformin on various cells have shown different IC50%. Additionally, the presence of metformin in the plasma led to its absorption by normal tissue due to the expression of the cell surface drug transporter OCT1 on its surface. On the other hand, OCT1 expression is varied between tumor cells, just as metformin concentration is varied between tumor cells due to vascularization. Therefore, the same clinical dose does not achieve the same concentration in different tumor models, and thus, some tumors show a weaker response compared to others. Therefore, determination of the optimal doses for different tumors is the main restriction of the clinical translation of metformin. This can be achieved by examination of OCT1 expression and concentration of metformin in several tumors in human studies. Moreover, dosage adjustments based on individual response, side effects, and health status can help optimize its clinical utility. However, the first official blinded clinical trial of metformin with a survival objective demonstrated no benefit despite plasma drug levels being observed in the micromolar range [71]. Hence, another issue that we should take into account is the mutational status of LKB1 and genes encoding mitochondrial complex I components. In conclusion, the intriguing possibility that metformin or novel biguanides may have a therapeutic role to play in the treatment of diseases other than diabetes requires further research in pharmacokinetics and mechanism.

Future directions for research could focus on further elucidating the molecular mechanisms by which metformin modulates the immune response in the tumor microenvironment. Understanding the precise pathways and signaling molecules involved in metformin’s effects on immune cells would provide valuable insights for developing more targeted and effective combination therapies. Metformin combination with other small molecule modulators for key signaling pathways could be a highly promising combination for cancer immunotherapy.

Moreover, investigating the efficacy of metformin in different cancer types and stages, as well as exploring its potential synergy with other immunotherapies or targeted therapies, would be important for optimizing treatment strategies. Clinical trials evaluating metformin in combination with ICIs or other immunotherapies could provide valuable evidence for its clinical utility.

Overall, the findings suggest that metformin holds promise as an adjunct therapy to enhance the antitumor immune response. Further research and clinical studies are needed to fully understand the mechanisms of action and determine the optimal treatment regimens for maximizing the benefits of metformin in cancer immunotherapy.

## Figures and Tables

**Figure 1 ijms-24-13353-f001:**
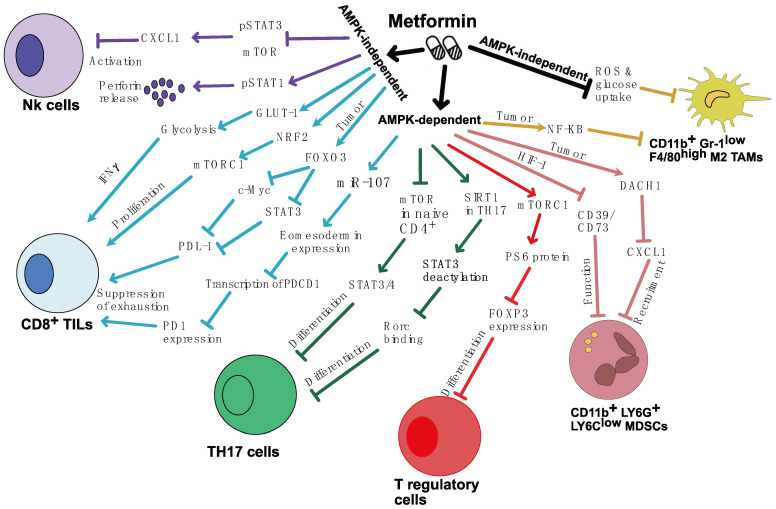
Effect of Metformin on tumor-infiltrated immune cells. Metformin can target several pathways in CD8+, NK cells, T regulatory cells, MDSCs, and TAMs in the tumor microenvironment in AMPK-dependent and AMPK-independent manners. AMPK: adenosine monophosphate-activated protein kinase; FOXO3: forkhead box O3; c-Myc: cellular Myc; STAT: signal transducer and activator of transcription; PD-L1: programmed death-ligand 1; PD-1: programmed cell death protein 1; NRF2 nuclear factor erythroid 2-related factor 2; mTORC1: mammalian target of rapamycin complex 1; GLUT-1: glucose transporter 1; IFNγ: interferon gama; CXCL1: C-X-C motif chemokine ligand 1; NK; natural killer; PS6: phosphorylation of S6 ribosomal protein; TH17: T helper 17; SIRT1: Situnin 1; Rorc: retinoic acid receptor-related orphan receptor; FOXP3: forkhead box P3; HIF-1: hypoxia-inducible factor-1; MDSCs: myeloid-derived suppressor cells; DACH1: dachshund family transcription factor 1; NF- κB: nuclear factor kappa B; TAMs: tumor-associated macrophages; ROS: reactive oxygen species.

**Table 1 ijms-24-13353-t001:** Effect of metformin on CD8^+^ TILs in different preclinical cancer models.

Cancer Model	Single or Combined	Dose	Effect on CD8^+^ TILs	Reference
Leukemia mouse model Melanoma mouse model	Single	5 mg/mL in the drinking water	Enhance effector and effector memory CD8^+^ TILs, accompanied with an increase in the cytokine production, including IL-2, IFNγ, and TNFα	[20,21]
Colon cancer	Single	250 mg/kg/ i.p	Increase CD8^+^ IFNγ^+^ TILs	[22]
Melanoma mouse model	Single	500 mg/kg orally	Increase CD8^+^CD69^+^, CD8^+^ IFNγ^+^, and CD8^+^ Granzyme B^+^ T cells as well as CD3^+^ effector-memory CD8^+^T cells (CD44^hi^ CD62L^lo^)	[23]
TNBC mouse model	Single	500 mg/kg orally	Increase CD8^+^ TILs Reduce exhausted CD8^+^ PD-1^+^ TILs	[33]
Breast cancer mouse model	Single	200 mg/kg/ i.p	Increase CD8^+^ TILs Increase granzyme B production from CD8^+^ TILs	[34]
Ovarian tumor mouse model	Single	200 mg/kg/ i.p	Increase CD8^+^ TILs	[35]
Melanoma mouse model Fibrosarcoma mouse model	Single and combined with anti-PD-1	5 mg/mL in the drinking water	Increase proliferation of CD8^+^ TILs	[24]
Triple-negative breast cancer (TNBC) mouse model	Combined with anti-PD-1	200 mg/kg/ i.p	Increase CD8^+^ TILs	[25]
Melanoma mouse model	Combined with anti-PD-1	100 mg/kg/ i.p	Increase CD8^+^ TILs Increase granzyme B production	[37]
Human papillomavirus-associated head and neck cancer (mEER/MTEC) mouse model	Combined with anti-PD-1	40 mg/kg/ i.p or 5 mg/mL in the drinking water	Increase CD8^+^ TILs infiltration and proliferation	[38]
Pancreatic cancer mouse model	Combined with HSV-1 oncolytic virus	5 mg/mL in the drinking water	Increase CD3^+^ CD8^+^ TILs infiltration and CD3^+^ CD8^+^ IFNγ^+^	[40]
Breast cancer mouse model	Combined with a tumor membrane vesicles (TMVs) vaccine	50 mg/kg/ i.p	Increase CD8^+^ TILs infiltration, accompanied by an increase in central memory CD8^+^ T cells, CD8^+^ IFNγ^+^ TNF-α^+^, and CD8^+^ IFNγ^+^ IL-2^+^	[43]

**Table 2 ijms-24-13353-t002:** Clinical studies of metformin targeting tumor-infiltrating immune cells in different types of cancer.

Cancer Type	Purpose	Clinical Trial Phase	Dose	Single or Combined	Effect on Tumor-Immune Cells	Status	Place	Number of Patients	Clinical Trial Identifier/Reference
HNSCC	To decrease TOMM20 expression in SCC and decrease MCT4 expression in fibroblasts	Early phase I	2000 mg/day	Single	Increase CD8^+^ TILs Decrease FOXP3^+^ cells Increase Nk cells	Completed	United States	50	NCT02083692/ [26,27,58]
ESCC	Enhance anticancer immunity	Phase II	250 mg/day	Single	Increase CD8^+^ TILs Decrease CD163^+^ macrophages Increases tumor-suppressive CD11c^+^ macrophages	Completed	China	128	ChiCTR-ICR-15005940/ [28]
CRC	To examine the pathological characteristics of resected CRC	NA	500–1000 mg/day	Single	Increase CD8^+^ TILs Decrease M2 macrophages	NA	Japan	267	[29]
Early breast cancer	To evaluate the effects of preoperative metformin in patients with stage I or IIA breast cancer	NA	500 mg/day (for 3 days) 750 mg/day (for 4 days) then 1000 mg/day (for 7 days)	Single	Increase CD8^+^ and CD4^+^TILs	NA	Japan	17	[30]
EOC	To study the impact of metformin on the TME of patients with EOC	Phase II	750 mg/day	Metformin plus platinum-based standard of care chemotherapy	Decrease CD4^+^ FOXP3^+^ cells	Completed	United States	82	HUM00047900/ [31]
NSCLC	To study the anticancer effects of metformin	NA	Not determined	Single	Increase central memory and memory stem CD8^+^ TILs	NA	China	58	[36]
Refractory/recurrent solid tumors (stage 1), advanced or recurrent NSCLC, advanced or recurrent thymic epithelial tumor, and advanced or recurrent pancreatic cancer (stage 2)	To investigate the safety, efficacy, and pharmacokinetics of a metformin–nivolumab combination treatment	Phase Ib	500–2250 mg/day	Metformin plus Nivolumab	Tumor-infiltrating immune cells	Ongoing	Japan	30	UMIN000028405/ [39]
Ovarian cancer	To investigate the antitumor efficacy of metformin on ovarian cancer patients	NA	Not determined	Single	Blocks the suppressive function of MDSC	NA	China	52	[64]
ESCC	To investigate the effect of metformin on MDSCs migration	NA	Not determined	Single	Decrease PMN-MDSCs accumulation	NA	China	75	[65]

HNSCC: head and neck squamous cell carcinoma; ESCC: esophageal squamous cell carcinoma; CRC: colorectal cancer; EOC: epithelial ovarian cancer; NSCLC: non-small cell lung cancer.

## Data Availability

Not applicable.

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
