# Peer review of "The Impact of Metformin on Tumor-Infiltrated Immune Cells: Preclinical and Clinical Studies"

_ijms, 2023, doi:10.3390/ijms241713353_

Round 1
Reviewer 1 Report
In this manuscript, the authors reviewed the effects of Metformin on tumor-infiltrated immune cells and its related signal pathways and discussed its potential therapeutic value in the clinic. Overall, it is a good and well-written review. I have a few minor comments and suggestions below:
1. The authors missed the important TIL populations, the T helper cells, such as Th1, Th17, et al. Although the author talked about Treg cells, they should separately discuss the role of Metformin in other T helper subsets.
2. Although the author comprehensively summarized the effects of Metformin on TILs, it lacks significant and deep discussion about its specific roles in different cell types.
3. Does Metformin have protumor effects? via regulating TILs? It seems that the authors did not touch on this aspect.
4. Figure 1 is too rough. I suggest the authors use appropriate tools, like Biorender, to improve this picture.
5. It would be great if the authors could provide several strategies that can enhance the clinical utility of Metformin.
It's OK if can improve further
Reviewer 2 Report
the manuscript is interesting and generally well written. Only minor revisions are required. See:
Line 44-45: It deserves to be added that another cause of immunotherapy treatment fail is the tumor vascularization since it plays a key role in tumor-infiltrating immune cells (see PMID: 37443812)
Lines 71-78: since this is a review article, It deserves to be added that metformin can also restore chemotherapy sensitivity modulating NF-kB, ERK1/2 activation, autophagy and cancer stem cells population (see PMID: 36361682)
Authors should add a table summarizing the main results found in the studies discussed
A schemathic representation of the effect of metformin on Tumor-Infiltrated Immune Cells should be added
